# Pathological Characteristics of the Emerging Recombinant African Swine Fever Virus Genotypes I and II in Vietnam

**DOI:** 10.3390/pathogens14090875

**Published:** 2025-09-02

**Authors:** Viet Dung Nguyen, The Viet Hoang Nguyen, Ngoc Duong Vu, Thi Tam Than, Thi Chau Giang Tran, Thi Thu Hang Vu, Thi Lan Nguyen, Yeon Hee Kim, Aruna Ambagala, Van Phan Le

**Affiliations:** 1Department of Microbiology and Infectious Disease, College of Veterinary Medicine, Vietnam National University of Agriculture, Hanoi 100000, Vietnam; nguyendungdhnl@gmail.com (V.D.N.); hoangvietnguyen1.vnua@gmail.com (T.V.H.N.); nguyenlan@vnua.edu.vn (T.L.N.); 2Faculty of Animal Science and Veterinary Medicine, Bac Giang Agriculture and Forestry University, Bac Giang 230000, Vietnam; 3Laboratory of Viral Infectious Diseases, Center for Research Excellence and Innovation, Vietnam National University of Agriculture, Hanoi 100000, Vietnam; duongvnd99@gmail.com (N.D.V.); thantam207@gmail.com (T.T.T.); chaugiangtran1205@gmail.com (T.C.G.T.); 4Institute of Veterinary Science and Technology, Vietnam Union of Science and Technology Association, Hanoi 100000, Vietnam; moonrtd@gmail.com; 5Foreign Animal Disease Division, Animal and Plant Quarantine Agency, Gimcheon 39660, Republic of Korea; vetyh@korea.kr; 6Canadian Food Inspection Agency, National Centre for Foreign Animal Disease, Winnipeg, MB R3E 3R2, Canada; 7Department of Medical Microbiology and Infectious Diseases, Max Rady College of Medicine, University of Manitoba, Winnipeg, MB R3E 0J9, Canada

**Keywords:** ASFV, clinical signs, pathology, recombinant ASFV genotype I/II strain

## Abstract

African swine fever (ASF) is a highly lethal disease caused by the ASF virus (ASFV) and poses a significant threat to the swine industry worldwide. This study investigated the pathogenicity and pathological characteristics of VNUA/rASFV/HD1/23, a recently identified recombinant ASFV genotype I/II in northern Vietnam. Sixteen healthy, seven-week-old pigs divided into four groups were inoculated intramuscularly (IM) with different virus concentrations (10^2^, 10^3^, and 10^4^ HAD_50_/mL), and their clinical signs, survival times, and pathological alterations were evaluated. All experimentally infected pigs exhibited acute clinical signs characterized by fever, anorexia, depression, diarrhea, and death (4–10 days after injection). The pathological findings included splenomegaly with infarcts, hemorrhagic lymph nodes, and severe pulmonary congestion. The pigs that received the highest dose (10^4^ HAD_50_/mL) IM showed the earliest onset of clinical signs and the shortest survival time. This study provides important insights into the virulence and the pathological lesions induced by the recombinant genotype I/II ASFV strains that emerged in Vietnam.

## 1. Introduction

African swine fever (ASF) is a devastating viral disease with a high mortality rate in both domestic and wild pigs. The causative agent, ASF virus (ASFV), is a DNA virus belonging to the genus *Asfivirus* and the family *Asfarviridae* [1]. The ASFV genome encodes more than 150 proteins, and the B646L gene, which encodes the p72 protein, is commonly used to classify ASFV strains into 24 different genotypes [2].

ASF was first observed in East Africa in 1907 and formally described in Kenya in 1921. ASF was introduced to Portugal in 1957 (ASFV genotype I) and to Georgia in 2007 (ASFV genotype II) [3]. The first reported ASF outbreak in Asia occurred in August 2018 in Liaoning Province, China, caused by a highly pathogenic ASFV genotype II strain [4]. In 2021, China reported the detection of low virulent genotype I ASFV strains genetically similar to naturally attenuated genotype I strains reported from Portugal [5]. In addition, several attenuated genotype II strains were reported from China [6]. In 2023, highly virulent recombinant ASFV strains of genotypes I and II (rASFV I/II) emerged in Jiangsu, Henan, and Inner Mongolia, which showed resistance to protection induced by HLJ/18- 7GD, a seven-gene strain [7]. Notably, similar recombinant ASFV strains were detected in domestic pigs in Russia near the Chinese border in 2023, highlighting the potential for cross-border transmission and the increasing genetic complexity of circulating ASFV isolates [8].

ASF was first reported in Vietnam in February 2019 in the province of Hung Yen and quickly spread nationwide. The virus responsible was identified as a highly pathogenic genotype II ASFV, which is closely related to the strains circulating in China at that time [9]. Later, a low virulent ASFV field isolate with deletions in the left region of the variable multigene family (MGF) was detected in Vietnam [10]. In 2023, several rASFV I/II strains were discovered during surveillance activities in six northern provinces of Vietnam [11]. These rASFV I/II strains are resistant to the two live-attenuated ASF vaccines licensed in Vietnam [12]. 

In this study, we investigated the pathogenicity and pathological lesions induced by VNUA/rASFV/HD1/23, one of the rASFV I/II strains isolated from Northern Vietnam [11]. Understanding the virulence and pathology of this strain provides important insights into the management of ASF outbreaks in Vietnam and the region.

## 2. Materials and Methods

### 2.1. Ethics Statements

The animal study was conducted in the biosafety level 3 facility at the College of Veterinary Medicine, Vietnam National University of Agriculture (VNUA), Hanoi, Vietnam, in compliance with the 2011 Guide for Care and Use of Laboratory Animals (eighth edition) and the guidelines of good experimental practices approved by the Committee on Animal Research and Ethics at the College of Veterinary Medicine, VNUA, Hanoi, Vietnam.

### 2.2. Virus Strain

The virus used in this study was a field-isolated rASFV I/II strain VNUA/rASFV/HD1/23 (GenBank Accession No. OR999147) [11], which was propagated in porcine primary alveolar macrophages (PAMs) collected from 8- to 9-week-old healthy piglets. To confirm the absence of common swine viruses in the PAM cells, real-time PCR assays (Median Diagnostics Inc., Chuncheon-si, Gangwon-do, South Korea) specific for porcine circovirus type 2 (PCV2), classical swine fever virus (CSFV), porcine reproductive and respiratory syndrome virus (PRRSV), and ASFV were used. PAM cells cultured in RPMI-1640 medium (Gibco) supplemented with 10% fetal bovine serum (FBS) and 1% antibiotics were seeded onto 96 tissue culture plates at a density of approximately 4 × 10^5^ cells/well. After 48 h of ASFV infection, 20 µL of 1% porcine red blood cells in RPMI medium was added to each well containing PAM cells. The formation of hemadsorption (HAD) rosettes on ASFV-infected PAM cells was observed daily for the next five days under an inverted microscope. The HAD titer of the virus was calculated according to the method of Reed and Muench, as previously described [13].

### 2.3. Animals

In this study, a total of 16 healthy, weaned seven-week-old, crossbred Yorkshire–Landrace–Duroc piglets were obtained from a high-health commercial pig farm in Hung Yen, Vietnam. All pigs were tested negative for ASFV, CSFV, PRRSV, FMDV, and PCV2 genomic material by real-time PCR (Median Diagnostics Inc., South Korea). Pigs were also tested negative for antibodies against ASFV by ELISA (VDPro® ASFV Ab i-ELISA ver 2.0 Kit, Median Diagnostics Inc., South Korea). The pigs were randomly assigned to four pens (four pigs per group) in the biosafety level 3 experimental facility at the College of Veterinary Medicine, Vietnam National University of Agriculture (VNUA), Hanoi, Vietnam. They had *ad libitum* access to feed and water and were observed and monitored daily. After a five-day acclimatization period, pigs in Groups 1, 2, and 3 were injected intramuscularly with ASFV at a dose of 10^2^, 10^3^, and 10^4^ HAD_50_/mL, per pig, respectively. The pigs in the control group were given 1 mL of sterile PBS intramuscularly per pig, and all pigs were monitored daily for clinical signs.

### 2.4. Sample Collection

Whole blood (EDTA) and serum samples were collected from each pig from the jugular vein starting on day 0, and then on 3-, 6-, and 9-day post-infection. Oral fluid samples were collected by hanging a cotton rope (TEGO Swine Oral Fluid kit, ITL Biomedical, Reston, VA, USA) in each pen daily for 30–45 min. The wet rope was squeezed into a clean plastic bag to collect the oral fluid. The oral fluid was aliquoted into 2.5 mL cryovials and frozen at −80 °C until analysis. Complete necropsies were immediately performed on all deceased/euthanized pigs, and tissue samples were taken from organs such as the spleen, liver, kidneys, tonsils, and lymph nodes for ASFV genome detection and histopathological examination. 

### 2.5. DNA Extraction and Real-Time PCR

Total DNA was extracted from whole blood and oral fluid samples using the QIAamp DNA Mini Kit (Qiagen, Hilden, Germany) according to the manufacturer’s instructions. Detection of rASFV I/II genomic material was performed using a previously reported real-time PCR assay [12]. All real-time PCR reactions were prepared using TaqMan® Fast 1-Step Master Mix (Thermo Fisher Scientific, Waltham, Massachusetts, USA) and performed using a CFX96 Touch Real-Time PCR Detection System (Bio-Rad Laboratories Ltd., Hercules, CA, USA).

### 2.6. Scoring of ASF Clinical Signs

A previously described clinical scoring system [3] was used to assess the clinical signs in VNUA/rASFV/HD1/23-infected pigs. For each criterion, a score was assigned for either normal (score 0), mildly altered (score 1), marked clinical symptom (score 2), or severe ASF symptom (score 3). The values for all parameters and the total score of the individual pigs were estimated daily. If a pig reached the total clinical score of 10 points, it was humanely euthanized as previously described [3].

## 3. Results

### 3.1. Clinical Signs

In the control group (non-infected pigs), all animals remained healthy throughout the experiment and showed no ASF-related clinical signs. In contrast, pigs in the ASFV-infected groups showed fever, anorexia, depression, diarrhea, and recumbency starting between 3 and 5 days post-infection (dpi) (Table 1). The most frequently observed clinical signs were fever and depression (100%), followed by anorexia (92%) and diarrhea (33%). The time of onset of fever was dose-dependent. In Group 3 (10^4^ HAD_50_), three out of four infected pigs showed elevated rectal temperatures at 3 dpi. After 4 dpi, all pigs in this group developed a fever that lasted until death. In Group 2 (10^3^ HAD_50_), only one of the four pigs developed fever at 3 dpi. At 4 dpi, another pig developed fever, and at 5 dpi, all pigs developed fever, which persisted until death. In Group 1 (10^2^ HAD_50_), one pig developed fever at 5 dpi, and after 6 dpi, all pigs developed fever (Table 1, Figure 1).

The other ASF-related clinical signs followed a similar pattern, and the average clinical score of Group 3 (10^4^ HAD_50_) started to increase on the third day. In Groups 1 and 2, the average clinical scores began to increase on 5 dpi. In all three ASFV-infected groups, the average clinical scores increased rapidly (Figure 2).

All ASFV-infected pigs succumbed to the infection between 4 and 10 days post-infection (dpi), and a difference in survival time was observed between the three ASFV-infected groups. Mortality occurred earliest in Group 3 (10^4^ HAD_50_), where one of four infected pigs was found dead on 4 dpi. On 5 dpi, two other pigs in Group 3 were euthanized as they reached the humane endpoint. The remaining pig in Group 3 reached the humane endpoint on 6 dpi. In Group 2 (10^3^ HAD_50_), two out of four pigs reached humane endpoint on 6 dpi, and therefore were euthanized. The remaining two pigs were euthanized on 7 and 8 dpi as they reached the human endpoint. In Group 1 (10^2^ HAD_50_), the first pig was euthanized on 6 dpi, and the remaining three pigs were euthanized on 7, 8, and 10 dpi as they reached humane endpoint (Table 1, Figure 3). To statistically analyze the survival differences, a log-rank test (Mantel–Cox) was performed to compare the survival curves of all four test groups. The analysis revealed a significant difference in the probability of survival (χ^2^ = 20.6, df = 3, *p* = 0.0001). The control group had no mortality, while Group 3 (10^4^ HAD_50_) had the fastest and most complete mortality, which contributed significantly to the overall test statistic. These results confirm that the survival rate was significantly influenced by the infectious ASFV dose.

### 3.2. Detection of ASFV Genomic Material in Whole Blood and Oral Fluid

Whole blood samples taken on 0, 3, 6, and 9 dpi were tested for ASFV genomic material by real-time PCR. The results showed that viremia was detected at 3 dpi in all pigs in experimental Groups 2 (10^3^ HAD_50_) and 3 (10^4^ HAD_50_). In Group 1 (10^2^ HAD_50_), two pigs developed viremia by 3 dpi. Viremia was detected in the remaining two pigs on 6 dpi. It is plausible that those two pigs developed viremia between 3 and 6 dpi (Table 1). 

Oral fluid samples were collected daily from all four groups. The samples could be obtained only up to 3 dpi from Group 3, up to 5 dpi. In Group 3 (10^4^ HAD_50_), ASFV was first detected at 2 days post-inoculation (dpi) with a Ct value of 35.59, indicating the onset of viral shedding. In Group 2 (10^3^ HAD_50_), the virus was not detected until 4 dpi (Ct = 37.41. In contrast, Group 1 (10^2^ HAD_50_) exhibited no detectable viral load until 5 dpi, with a Ct value of 37.4, indicating a delayed onset of viral replication and shedding in animals inoculated with the lowest infectious dose. No viral genome was detected in the control group throughout the observation period, confirming the validity of the negative control (Table 2).

These findings demonstrate a clear dose-dependent pattern in the timing of ASFV detection in oral fluids, with higher infectious doses leading to earlier detection.

### 3.3. Gross Pathological Findings

Gross lesions observed in all pigs included hemorrhagic and enlarged lymph nodes, tonsillar erythema, and splenomegaly. Renal lesions, including petechiae on the renal surface and hemorrhages in the renal pelvis, were observed in 11 of 12 pigs, except pig No. 15. The pathologic lesions observed in some pigs included pneumonia (11/12) and petechiae on the serosa (7/12). The gross lesions in individual pigs are summarized in Table 3 and are shown in Figure 4.

The spleen was significantly larger and dark red with infarcts in all pigs in the three infection groups (Figure 4A, red arrow). The tonsil, mandibular, and inguinal lymph nodes were enlarged, hemorrhagic, and/or congested (Figure 4D,F,I). The mesenteric lymph nodes were also markedly congested and enlarged (Figure 4B, red circle). The lungs were congested, hemorrhagic, and edematous, with moderate to severe interstitial pneumonia (Figure 4C). The kidneys were damaged with petechiae on the surface of the kidneys and hemorrhage in the renal pelvis (Figure 4E, green circle and Figure 4H). Hyperemia in the gastric mucosa was observed in 11/12 pigs (Figure 4G).

### 3.4. Histopathological Lesions

The main histopathological lesions observed in the experimental pigs were severe vascular disorders and inflammatory reactions in several organs. In the spleen, extensive vascular congestion and hemorrhages led to obliteration of the white pulp architecture. In the splenic parenchyma, scattered small fibrinoid necrotic foci were interspersed with apoptotic cells, while the stromal tissue showed considerable neutrophil infiltration (Figure 5(A2,A3)).

The lymph nodes showed diffuse, moderate to severe hyperemia, with the sinuses infiltrated by mononuclear cells, pyknotic cells undergoing karyorrhexis, and cellular debris. This was associated with moderate to severe lymphocyte depletion in lymphoid follicles and interfollicular areas, often in association with numerous pyknotic cells (Figure 5(B2,B3)). In contrast, moderate lymphocyte depletion was observed in the palatine tonsil, affecting both lymphoid follicles and interfollicular areas. This depletion was accompanied by the presence of pyknotic cells, cellular debris, and the infiltration of villous macrophages, the latter being particularly abundant in the lymphoid follicles. The crypt epithelium typically showed moderate to heavy infiltration of mononuclear cells characterized by pyknosis and cell fragmentation (Figure 5(C2,C3)).

In the lungs, the perialveolar blood vessels showed severe congestion, with erythrocytes extravasating into the perialveolar interstitium in certain areas. Some regions showed edema with serous exudate accumulating in the alveolar lumen, while the intralobular septa between the alveoli was infiltrated by neutrophil, lymphocytes and scattered histiocytes, suggesting an inflammatory reaction (Figure 5(D2,D3)).

The gastric mucosa showed scattered degeneration and desquamation of the epithelial layer with marked congestion and dilation of blood vessels in the mucosal and submucosal layers, leading to the luminal accumulation of erythrocytes. The stromal tissue was highly edematous and infiltrated by lymphocytes and neutrophil granulocytes, which agglomerated into several small clusters in the mucosal layer (Figure 5(E2,E3)).

The congestion and dilation of blood vessels were also noted in the kidneys, both in the renal cortex and medulla, with numerous erythrocytes within the vascular lumina. Some cortical blood vessels also showed mild hemorrhage, while scattered small clusters of renal tubule cells showed mild degeneration (Figure 5(F2,F3)).

## 4. Discussion

African swine fever (ASF) has caused considerable economic losses to the Vietnamese swine industry. The disease has become endemic in Vietnam, with several virus strains of varying virulence circulating simultaneously. These include highly virulent ASFV p72 genotype II strains [9,15,16], low virulent gene-deleting ASFV p72 genotype II strains [10], and, most importantly, the emergence of highly virulent recombinant ASFV genotype I and II strains (rASFV I/II) [12,17], complicating the ASF control measures in Vietnam. Although the two commercial ASFV vaccines currently licensed and marketed in Vietnam have been shown to protect pigs against highly virulent ASFV p72 genotype II strains isolated in the country [18], they do not protect against rASFV I/II strains [12]. Considering the variable pathogenicity of ASFV, ranging from acute to chronic disease depending on the viral strain, it is crucial to investigate the pathological characteristics of pigs experimentally infected with strains isolated in the field. 

This study aimed to investigate the virulence of a rASFV I/II strain of VNUA/rASFV/HD1/23 in experimentally infected pigs using different doses. To our knowledge, this is the first experimental infection study conducted in Vietnam to evaluate the virulence of a rASFV I/II strain isolated in the country. In pigs infected intramuscularly with the VNUA/rASFV/HD1/23 strain, clinical signs such as pyrexia, anorexia, depression, recumbency, and death occurred between 4 and 10 days post-infection (dpi). Diarrhea was observed in some animals, consistent with the findings of a previous study [19]. The ASFV genome was first detected in the blood of most pigs of the three experimental groups 3 days post-infection (dpi), i.e., 1–2 days before the onset of fever. This observation is consistent with the timing of viremia reported for the ASFV strains involved in the outbreaks in Korea between 2022 and early 2023 [20], as well as with that observed for the rASFV I/II strains detected in China and Russia [7,8]. Interestingly, ASFV was detected in Group 3 as early as 2 dpi in oral fluid, one day before detection in blood. In contrast, detection in oral fluid in Groups 2 and 1 occurred later, at 4 dpi and 5 dpi, respectively. The earlier detection of ASFV in oral fluid compared to blood in Group 3 was most likely due to the sampling strategy, as blood samples were only collected on day 3. In contrast, oral fluid samples were collected daily. Previous experimental studies have shown that the detection of ASFV in blood and oral fluids can take between 2 and 19 days, with the duration influenced by factors such as the pathogenicity of the virus strain, infectious dose, route of infection, and host characteristics [7,8,14,15,20]. 

Pathological findings at postmortems showed the typical lesions of acute ASFV, such as splenomegaly with congestion and hemorrhagic lymphadenopathy. The histopathological appearance of the spleen will include vascular congestion, and hemorrhage resulted in the destruction of the white pulp structure. Small fibrinoid necrotic foci were found among apoptotic cells in the splenic parenchyma, with notable neutrophilic infiltration in the stromal tissue [21,22]. The most impacted lymphatic nodes exhibited moderate to severe hyperemia, characterized by sinuses permeated by mononuclear cells, pyknotic cells undergoing karyorrhexis, and cellular detritus. Histological lesions in the liver, lungs, and stomach were markedly analogous in other histopathological investigations involving various highly virulent ASFV strains [23,24]. 

The clinical picture, survival time, mortality rate, and pathological lesions observed in animals infected with VNUA/rASFV/HD1/23 closely resemble the acute form of ASFV. The disease progression of this strain is consistent with that of highly virulent ASFV p72 genotype II strains, including Georgia 2007/1 (isolated in Georgia, 2007), Pig/Heilongjiang/2018 (isolated in China, 2018), VNUA/HY/ASF1 (isolated in Vietnam, 2019), Mongolia/2019 (isolated in Mongolia, 2019), and Korea/Pig/Paju1/2019 (isolated in Korea, 2019). These strains are characterized by rapid progression, high mortality, and severe pathology [4,14,25,26,27]. The present study reconfirms the severe and acute pathology associated with highly virulent ASFV strains. It emphasizes the critical importance of early and accurate diagnosis for effective management in affected herds. Previous experimental studies have documented that the clinical course of ASF ranges from 3 to 19 days, depending on various factors, including the pathogenicity of the virus isolates, dose, route of infection, and host characteristics [27,28,29,30]. In the present study, the incubation period in pigs injected intramuscularly with 1 mL of the VNUA/rASFV/HD1/23 strain at doses of 10^4^, 10^3^, and 10^2^ HAD_50_ was 3 to 6 dpi, depending on the dose (Table 1). These results are comparable to the incubation times previously reported for the ASFV Pig/HLJ/18 (3–5 dpi) and HB31A (4–5 dpi) strains isolated in China in 2018 and 2020, respectively, the Georgia 2007/1 strain (3–6 dpi) isolated in Georgia in 2007, Mongolia/2019 (3–5 dpi) isolated in Mongolia in 2019, Korea/Pig/Paju1/2019 (4–6 dpi) isolated in Korea in 2019, and rASFV I/II strain of Primorsky/2023/DP/4560.Rec (3–6 dpi) isolated in Russia [4,8,19,25,26,27].

Previous studies conducted in Vietnam have shown that the mean death time of ASFV strains depended on the infection route. The mean death time of pigs infected intramuscularly with 10^3^ HAD_50_ was 7.0 ± 1.2 dpi [14]. In contrast, pigs infected orally with the same virus and dose had a slower disease onset and a longer disease course. None of the animals died suddenly, and the mean death time was 19.8 ± 4.66 days [15]. In this study, the mean death time of pigs after intramuscular inoculation of 1 mL of the VNUA/rASFV/HD1/23 strain at a dose of 10^4^ HAD_50_ was 5 ± 0.82 days, while with an intramuscular injection of 1 mL of the virus at a dose of 10^3^ and 10^2^ HAD_50_, the mean death time of was 6.75 ± 0.96 and 7.75 ± 1.71 days, respectively. The obtained mean death times in this study are comparable to those of the rASFV I/II strain of Primorsky/2023/DP/4560.Rec isolated in Russia (5.8 ± 0.9 dpi) [8], as well as the ASFV p72 genotype II strains of Pig/Heilongjiang/2018 (8.0 ± 1.4 dpi) [31], Korea/Pig/Paju1/2019 (8.0 ± 1.2 dpi) [27], Mongolia/2019 (6.5±0.8 dpi) [26], and Georgia 2007/1 (8.8 ± 1.1 dpi) [32]. The slight variation observed in the mean death times between the ASFV strains was likely due to the difference in the dose of the virus, route of inoculation, the age, breed, and the immune status of the pigs used in different studies. 

These results indicate that VNUA/rASFV/HD1/23 retains highly pathogenic characteristics comparable to global highly virulent ASFV p72 genotype II strains, with rapid disease progression posing challenges for containment. Importantly, the study found no significant differences in pathogenicity between the rASFV I/II strain of VNUA/rASFV/HD1/23 and the earlier ASFV p72 genotype II of the VNUA/HY/Vietnam strain isolated in 2019 in Vietnam. The mean death time of pigs infected intramuscularly with VNUA/HY/Vietnam at a dose of 10^3^ HAD_50_ was 7.0 ± 1.2 dpi [14], while the mean death time of pigs infected intramuscularly with VNUA/rASFV/HD1/23 at a dose of 10^3^ HAD_50_ was 6.75 ± 0.96. This finding underscores the ongoing threat of highly virulent ASFV strains circulating in Vietnam. The recombinant ASFV strain VNUA/rASFV/HD1/23 caused rapid disease progression and a high mortality rate, emphasizing the variability of ASFV virulence, even among strains circulating in the same geographic area. 

The present findings underscore the urgent need for the development of rapid diagnostic tools to detect highly virulent ASFV strains like VNUA/rASFV/HD1/23. Early detection is crucial for prompt intervention to minimize transmission and losses within farms. While intramuscular inoculation is a standard method for experimental studies, natural transmission routes, such as oral exposure or contact with the pathogen, can lead to different clinical outcomes. Further experimental studies on these transmission routes are essential for the development of effective biosecurity measures and vaccination strategies. The recombinant rASFV I/II strains pose a significant threat, as the two vaccines currently available in Vietnam are ineffective against them. Therefore, research into the prevalence and spread of these recombinant viruses is crucial for the development of targeted prevention and control measures that will ultimately mitigate the impact of ASF on the pig industry. Finally, the comparable pathogenicity and incubation periods of VNUA/rASFV/HD1/23 with other globally virulent ASFV strains highlight the critical need for international collaboration in ASF research. Joint efforts should focus on vaccine development, understanding strain-specific variations, and implementing targeted countermeasures to mitigate the impact of emerging recombinant ASFV on the global swine industry.

## 5. Conclusions

This study provides important insights into the pathogenicity and pathological characteristics of ASFV strain VNUA/rASFV/HD1/23, a new emerging recombinant ASFV strain of p72 genotypes I and II (rASFV I/II) currently circulating in northern Vietnam. The results of experimental animal infections in domestic pigs showed that this strain has a high virulence characterized by rapid disease progression, acute clinical symptoms such as fever, anorexia, and depression, and a high mortality rate within 4–10 days after infection. The pathological findings showed the typical lesions of acute ASF, such as splenomegaly with hyperemia, hemorrhagic lymphadenopathy, and severe pulmonary congestion. The observed dose-dependent severity of the disease emphasizes the need to understand the viral dynamics in order to develop effective prevention strategies. In particular, the VNUA/rASFV/HD1/23 strain showed pathogenicity comparable to highly virulent ASFV p72 genotype II strains, posing a significant challenge to existing control measures, including vaccines that are ineffective against rASFV I/II strains. These findings underscore the urgent need for improved diagnostic tools, stringent biosecurity measures, and continuous surveillance to address the increasing threat of ASFV.

## Figures and Tables

**Figure 1 pathogens-14-00875-f001:**
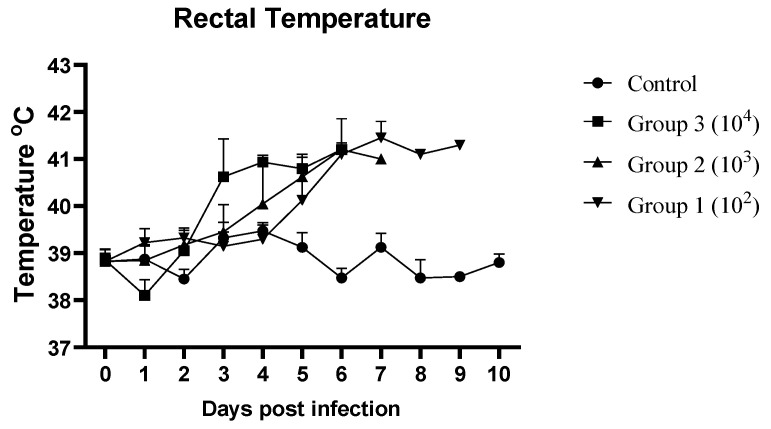
Mean daily rectal temperatures (°C) of ASFV-infected and non-infected (control) pig groups.

**Figure 2 pathogens-14-00875-f002:**
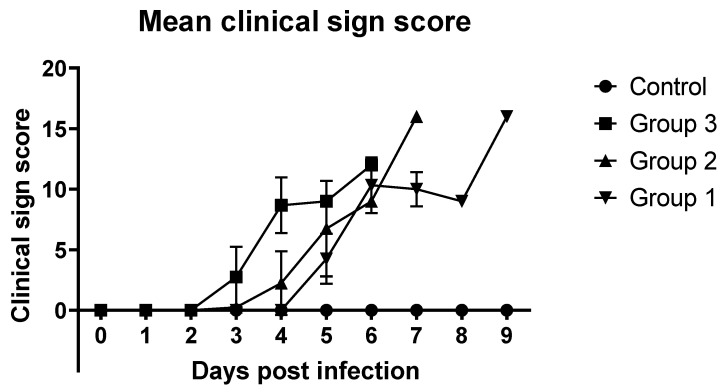
Mean clinical scores of ASFV-infected and non-infected groups of pigs. Scores were calculated as previously described [3,14].

**Figure 3 pathogens-14-00875-f003:**
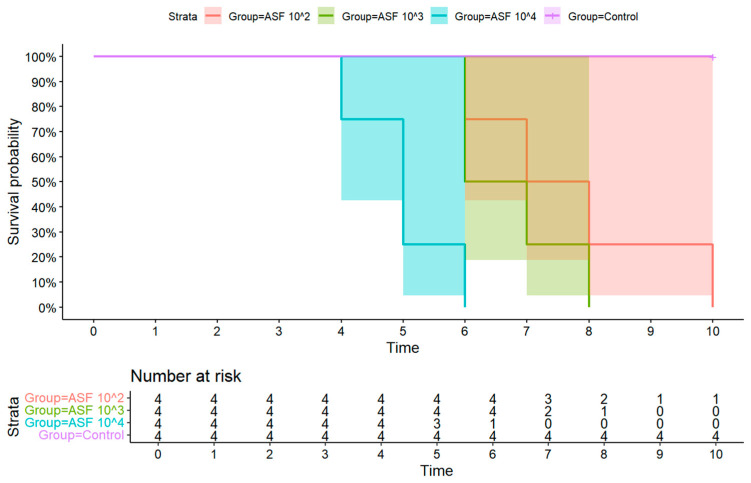
The survival rates of the control group (purple), Group 3 (green), Group 2 (light blue), and Group 1 (red). The figure was created using the programming language R (version 4.3.2).

**Figure 4 pathogens-14-00875-f004:**
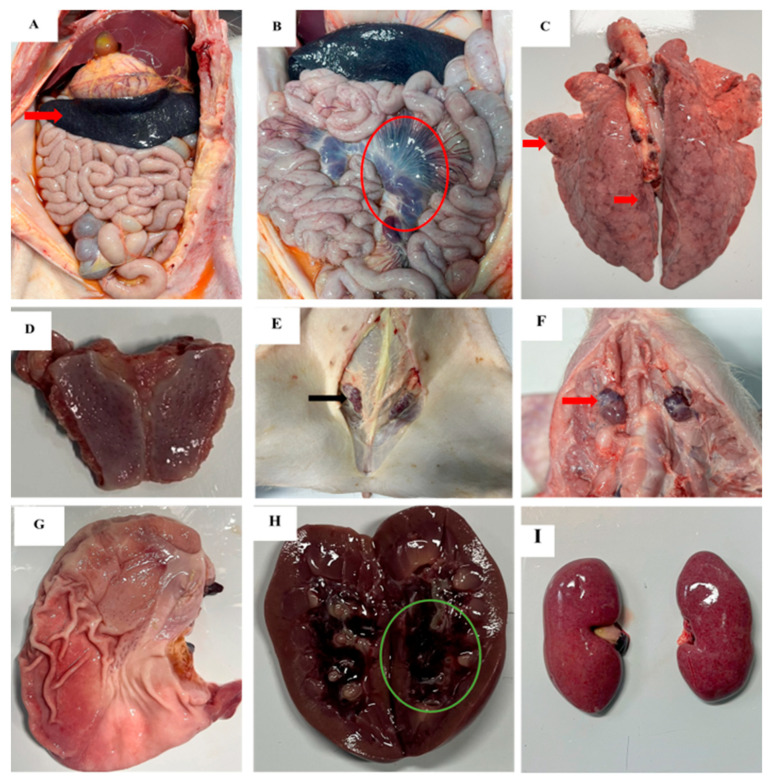
Gross pathologic lesions observed in pigs infected with VNUA/rASFV/HD1/23: (**A**) infarcts (red circle) and enlarged spleen; (**B**) hemorrhagic mesenteric lymph nodes (red circle); (**C**) pulmonary inflammation and edema; (**D**) hyperemic tonsils; (**E**) enlarged and congested inguinal lymph nodes (black arrow); (**F**) Enlarged and congested mandibular lymph nodes (red arrow); (**G**) congestion in the gastric mucosa; (**H**) hemorrhage in the renal pelvis (green circle); (**I**) numerous petechiae on the cortical surface of kidney.

**Figure 5 pathogens-14-00875-f005:**
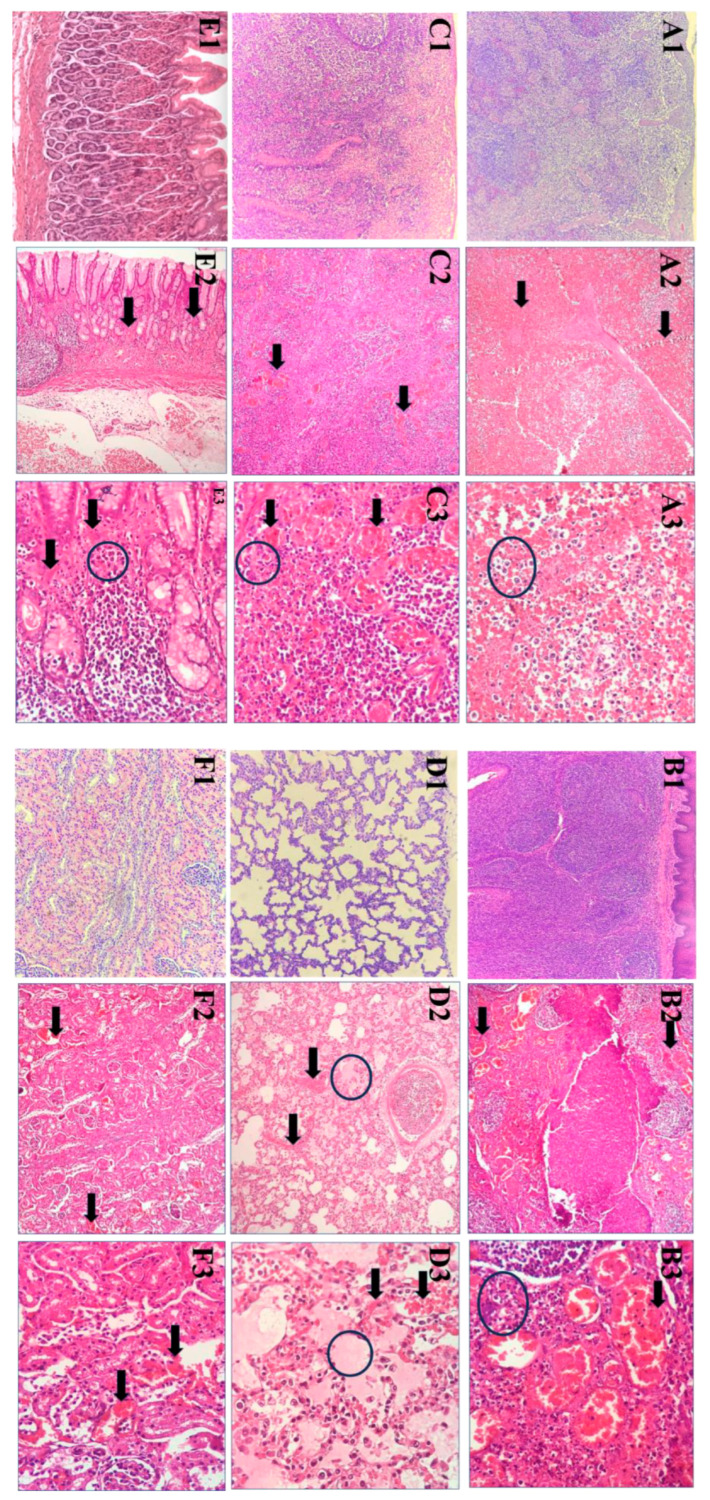
Histopathologic lesions (**A1**–**F1**) are the microscopic control specimens of the spleen, tonsils, inguinal lymph nodes, lung, stomach and kidney, respectively. (**A2**) The red pulp of the spleen shows severe and diffuse congestion (black arrows) (H&E 10×). (**A3**) Significant infiltration of hypertrophic and necrotic mononuclear cells, together with the presence of megakaryocytes (black circle) (H&E 40×). (**B2**,**C2**) The tonsils and inguinal lymph nodes show congestion with abundant red blood cells (black arrows) (H&E 10×). (**B3**,**C3**) Infiltrated by mononuclear cells and pyknotic cells (black circle) (H&E 40×) in the tonsils and inguinal lymph nodes. (**D2** (10×),**D3** (40×)) The lungs show severe congestion (black arrow) and edema (black circle) with fluid accumulation in the alveolar lumen. (**E2** (10×),**E3** (40×)) The gastric mucosa showed a luminal accumulation of erythrocytes (black arrow). The stromal tissue was highly edematous and infiltrated with lymphocytes and neutrophils (black arrow). (**F2** (10×),**F3** (40×)) Congestion and dilation of the blood vessels in both the renal cortex and the renal medulla with numerous erythrocytes in their lumina (black arrow).

**Table 1 pathogens-14-00875-t001:** Date (given in dpi) of onset of clinical symptoms and viremia. Fever was defined as a rectal temperature above 40 °C for more than two days.

Group	Pig No.	Date of Onset of Clinical Signs in Pigs After ASFV Infection						
		Fever	Depression	Anorexia	Diarrhea	Recumbency	Dead	Onset of Viremia (Ct Value)
Group 3 (10^4^ HAD_50_)	5	3	4	5	5	5	6	3 (21.36)
	6	3	3	3		3	4	3 (18.41)
	7	3	4	3		4	5	3 (18.95)
	8	4	4	4		4	5	3 (21.15)
Group 2 (10^3^ HAD_50_)	9	5	5	5		5	6	3 (23.67)
	10	3	5	5		5	6	3 (18.46)
	11	5	5	3	7	7	8	3 (33.96)
	12	4	5	5	6	6	7	3 (29.38)
Group 1 (10^2^ HAD_50_)	13	5	5	6		6	6	3 (30.4)
	14	5	5	6		6	7	3 (33.18)
	15	6	6	6	8	9	10	6 (18.36)
	16	5	5	6		7	8	6 (15.34)

Note: Ct (cycle threshold) refers to the number of PCR cycles required for the fluorescence signal to exceed the background value. Interpretation of the Ct value: Ct < 38: Positive; 38 ≤ Ct < 40: Suspicious; Ct ≥ 40: Negative.

**Table 2 pathogens-14-00875-t002:** Detection of ASFV genomic material in oral fluids. NS = No sample was recovered as pigs were not interested in chewing the ropes. (-) = Oral fluid samples tested negative for ASFV genomic material.

Dpi	Viral Load (Ct Value) of the ASFV in Oral Fluids			
	Control Group	Group 3 (10^4^ HAD_50_)	Group 2 (10^3^ HAD_50_)	Group 1 (10^2^ HAD_50_)
0	-	-	-	-
1	-	-	-	-
2	-	35.59	-	-
3	-	39.74	-	-
4	-	NS	37.41	-
5	-	NS	33.84	37.4

**Table 3 pathogens-14-00875-t003:** Summary of gross lesions in pigs inoculated with VNUA/rASFV/HD1/23 strain.

Gross Lesions		Group 3 (10^4^ HAD_50_)				Group 2 (10^3^ HAD_50_)				Group 1 (10^2^ HAD_50_)				Total Frequency
		5	6	7	8	9	10	11	12	13	14	15	16	
Lymph nodes	Enlargement	+	+	+	+	+	+	+	+	+	+	+	+	12/12 (100%)
	Hemorrhage	+	+	+	+	+	+	+	+	+	+	+	+	12/12 (100%)
Spleen	Enlargement	+	+	+	+	+	+	+	+	+	+	+	+	12/12 (100%)
	Congestion	+	+	+	+	+	+	+	+	+	+	+	+	12/12 (100%)
Tonsils	Erythema	+	+	+	+	+	+	+	+	+	+	+	+	12/12 (100%)
Lung	Pneumonia	+	+	+	+	+	+	+	+	+	+	+	-	11/12 (92%)
Kidneys	Hemorrhage	+	+	+	+	+	+	+	+	+	+	-	+	11/12 (92%)
Stomach	Hyperemia	+	+	+	-	+	+	+	+	+	+	+	+	11/12 (92%)
Intestine	Hemorrhage	-	+	+	-	-	+	+	+	-	+	+	-	7/12 (58%)

## Data Availability

The data presented in this study are available on request from the corresponding author.

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
