# Peer review of "Pathological Characteristics of the Emerging Recombinant African Swine Fever Virus Genotypes I and II in Vietnam"

_pathogens, 2025, doi:10.3390/pathogens14090875_

Round 1
Reviewer 1 Report
Comments and Suggestions for Authors
Review: "Pathological Characteristics of the Emerging Recombinant African Swine Fever Virus Genotypes I and II in Vietnam"
This manuscript authored by Viet Dung Nguyen et al. addresses the pathological findings in pigs infected with the emerging recombinant ASFV genotype I/II VNUA/rASFV/HD1/23 in Vietnam. The topic is relevant, especially in the context of evolving virus strains with altered virulence, but the study has several major technical shortcomings that need to be revised as follows:
Major points
Inconsistent/unspecific technical terminology
In the manuscript, the use of technical terms is inconsistent. Further, the authors switch between descriptive terminology and diagnostic technical terminology. For example, the authors describe a “clearly red lymph node”. Instead, appropriate terms such as “congested” or “hemorrhagic” should be used. However, terms must be applied specifically depending on the lesion.
- Descriptions such as “diffuse lymphoid tissue” should be replaced by commonly used morphological terms.
- The statement about “erythrocytes in the lumina” of the kidney is unclear. Which lumen are you referring to? Please by as specific as you can.
- Terms like “interstitial tissue” in the lungs are non-specific; please rephrase to “inter/intra-lobular septa”.
- The composition of inflammatory infiltrates (e.g., presence or absence of histiocytes in addition to neutrophils and lymphocytes, e.g. line 219) should be specified.
Missing presentation of results
Several important findings are described, but the methodological approach remains unclear or the results are not presented visually.
- Splenomegaly is mentioned, but the method of assessment (e.g. weight/size in relation to body weight) is not stated.
- A picture of splenic infarction is not included, although this is highlighted as a major lesion upon infection with VNUA/rASFV/HD1/23.
- Gross diagnosis of interstitial pneumonia is not possible; confirmation requires histology.
- Hemorrhages in lymph nodes are described grossly, but not supported by histology (only hyperemia).
- In several figures, important details are either unlabeled of low quality or do not illustrate the described lesion.
- Figure 4:
- Panel C: Unclear what is depicted here. Please indicate the lesion by using an arrow.
- Panel E: Not a suitable representation of a hemorrhagic lymph node. Please provide a cross section of a lymph node with higher magnification.
- Panel H: Poor image quality; needs better lighting.
- Panel I: Petechiae are barely visible.
- There are several discrepancies between gross and microscopic findings (e.g., lymph node hemorrhages not confirmed histologically) that should be revised.
- Figure 4:
Absence of lesion scoring and quantification
The study is solely descriptive and no semi-quantitative scoring of gross or histopathological lesions is provided. Scoring systems for ASFV induced lesions (both gross and microscopic) are well established in the literature (e.g., Galindo-Cardiel et al., 2013) and should be used for objective comparisons. Why did the authors refrain from semi-quantitative analysis?
Minor points
Line 25: Please replace “pathological lesions” with an appropriate term, e.g. pathological alteration. A lesion is pathological by definition.
Figures 1 & 2: Data points are hard to distinguish; color-coding individual groups would improve readability.
Figure 5: The references to panels are unclear in the main text; organs should be clearly mentioned, e.g. tonsil in C2 and C3. In the main text, only the lymph node is described.
Conclusion
The manuscript primarily exhibits technical weaknesses, hence, I recommend a major revision for it.
Consistent and correct use of pathological terminology and the appropriate presentation of key findings are needed.
The absence of a standardized scoring system for lesions results in a superficial description, which limits the scientific impact of the work. After all technical and terminological issues are adequately revised or responded, the manuscript can be further considered in Pathogens.
Author Response
- Reviewer’s Comment: The study is solely descriptive and no semi-quantitative scoring of gross or histopathological lesions is provided. Scoring systems for ASFV-induced lesions (both gross and microscopic) are well established in the literature (e.g., Galindo-Cardiel et al., 2013) and should be used for objective comparisons. Why did the authors refrain from semi-quantitative analysis?
Author’s answer: We thank the reviewer for this valuable comment. In our study, the semi-quantitative scoring system described by Galindo-Cardiel et al. (2013) was indeed applied to monitor disease progression and determine humane endpoints for euthanasia of infected pigs. However, we did not include the detailed semi-quantitative scoring data for gross and histopathological lesions in this manuscript, as these analyses will be presented in a subsequent publication together with immunohistochemical (IHC) findings. We believe that integrating lesion scoring with IHC results will enable a more comprehensive and objective evaluation of ASFV-induced pathology and will avoid fragmenting the presentation of these related datasets across separate reports.
- Reviewer’s Comment: Line 25: Please replace “pathological lesions” with an appropriate term, e.g. pathological alteration. A lesion is pathological by definition.
Author’s answer: We appreciate the reviewer’s suggestion. As recommended, we have replaced “pathological lesions” with “pathological alterations” at line 25 to avoid redundancy.
- Reviewer’s Comment: Figures 1 & 2: Data points are hard to distinguish; color-coding individual groups would improve readability.
Author’s answer: Thank you for your valuable suggestion. Whilst we understand the importance of clear group differentiation, we would like to point out that in Figures 1 and 2, clear marker shapes (e.g., squares, circles, triangles) are already used to represent the different experimental groups. This design ensures a clear visual separation, even in greyscale printing and for readers with colour vision deficiencies. Therefore, we believe that the current illustration format effectively conveys the group differences without the need for additional colour coding.
- Reviewer’s Comment: Figure 5: The references to panels are unclear in the main text; organs should be clearly mentioned, e.g. tonsil in C2 and C3. In the main text, only the lymph node is described.
Author’s answer: We thank the reviewer for this insightful comment. The following sentence has been added (Lines 227–232): “In contrast, moderate lymphocyte depletion was observed in the palatine tonsil, affecting both lymphoid follicles and interfollicular areas. This depletion was accompanied by the presence of pyknotic cells, cellular debris, and infiltration of villous macrophages, the latter being particularly abundant in the lymphoid follicles. The crypt epithelium typically showed moderate to heavy infiltration of mononuclear cells characterized by pyknosis and cell fragmentation (Figure 5C2, C3).”
- Reviewer’s Comment: In the manuscript, the use of technical terms is inconsistent. Further, the authors switch between descriptive terminology and diagnostic technical terminology. For example, the authors describe a “clearly red lymph node”. Instead, appropriate terms such as “congested” or “hemorrhagic” should be used. However, terms must be applied specifically depending on the lesion.
Author’s answer:
Original text (Line 202-203): The mesenteric lymph nodes were also clearly red and enlarged.
Edited text (Line 202-203): “The mesenteric lymph nodes were also markedly congested and enlarged”
Reviewer’s Comment: Descriptions such as “diffuse lymphoid tissue” should be replaced by commonly used morphological terms.
Author’s answer: We appreciate the reviewer’s suggestion. As recommended, we have replaced “diffuse lymphoid tissue” with “interfollicular areas” at line 226.
- Reviewer’s Comment: The statement about “erythrocytes in the lumina” of the kidney is unclear. Which lumen are you referring to? Please by as specific as you can.
Author’s answer: We appreciate the reviewer’s suggestion. As recommended, the phrase “…with numerous erythrocytes in the lumina” was replaced with “…with numerous erythrocytes within the vascular lumina” at line 244.
- Reviewer’s Comment: Terms like “interstitial tissue” in the lungs are non-specific; please rephrase to “inter/intra-lobular septa”.
Author’s answer: We appreciate the reviewer’s suggestion. As recommended, we have replaced “interstitial tissue” with “intralobular septa” at lines 235–236.
9. Reviewer’s Comment: Splenomegaly is mentioned, but the method of assessment (e.g. weight/size in relation to body weight) is not stated.
Author’s answer: We appreciate the reviewer’s comment regarding the assessment of splenomegaly. In this study, splenomegaly was identified through gross morphological evaluation during necropsy. In all infected animals, the spleens were markedly enlarged, dark red, and turgid compared with those of the controls. Although quantitative measurements (e.g., spleen-to-body weight ratio) were not recorded, the macroscopic changes were consistent and sufficiently pronounced to allow reliable identification of splenomegaly. In future studies, we will incorporate objective morphometric data to strengthen the assessment.
- Reviewer’s Comment: A picture of splenic infarction is not included, although this is highlighted as a major lesion upon infection with VNUA/rASFV/HD1/23.
Author’s answer: We thank the reviewer for this valuable comment. We acknowledge that the term “splenic infarction” was mistakenly used in the initial version of the manuscript due to our limited experience with English veterinary pathology terminology. After re-evaluating the gross and histopathological findings, we determined that the lesions observed in the spleen were more accurately described as splenic congestion rather than infarction. Accordingly, we have revised the manuscript and replaced all references to “infarction” with “congestion” to ensure accurate pathological descriptions. We appreciate the reviewer’s understanding.
- Reviewer’s Comment: Gross diagnosis of interstitial pneumonia is not possible; confirmation requires histology.
Author’s answer: We agree with the reviewer that a gross diagnosis of interstitial pneumonia is not appropriate, as this condition requires histopathological confirmation. Accordingly, we have revised the manuscript and replaced the term “interstitial pneumonia” (line 200) with the more general, grossly observable description “pulmonary inflammation and edema” (line 212). We thank the reviewer for this valuable clarification
12. Reviewer’s Comment: Figure 4:
- Panel C: Unclear what is depicted here. Please indicate the lesion by using an arrow.
- Panel E: Not a suitable representation of a hemorrhagic lymph node. Please provide a cross section of a lymph node with higher magnification.
- Panel H: Poor image quality; needs better lighting.
- Panel I: Petechiae are barely visible.
Author’s answer:
- Panel C: We thank the reviewer for pointing this out. We agree that the lesion in Panel C was not clearly indicated in the previous version. The figure has been revised, and an arrow has been added to clearly highlight the lesion of interest.
- Panel E: We appreciate the reviewer’s comment and agree that the image does not adequately represent a hemorrhagic lymph node. Upon reevaluation, we determined that the lesion corresponds more accurately to vascular congestion rather than hemorrhage. Accordingly, the figure legend and related text in the manuscript have been revised to describe the lymph node as congested instead of hemorrhagic.
- Panels H & I: We appreciate the reviewer’s observations regarding the quality of Panel H and the visibility of petechiae in Panel I. We acknowledge that the image in Panel H appears dim due to suboptimal lighting, and that the petechial lesions in Panel I are subtle. While the current resolution limits further enhancement, these lesions become more discernible when using the zoom function in the digital version of the manuscript (e.g., Word or PDF). We will consider re-capturing the images with improved lighting in future studies to ensure better clarity
Reviewer 2 Report
Comments and Suggestions for Authors
In the manuscript entitled "Pathological Characteristics of the Emerging Recombinant African Swine Fever Virus Genotypes I and II in Vietnam", Dr. Nguyen et al., describe the pathogenesis and biological proporties if ASFV recombinant strain (I/II genotype) VNUA/rASFV/HD1/23. The authors provided a good introduction, with adequate methods and results that cover all the biological proporties of the strain.
This work gives a good insight on the new recombinant strains that are circulating in Asia and Russia, which should be taken into consideration when developing new vaccines.
I have some comments.
- In the introduction (line 48) and discussion, please mention the recombinant virus that was also detected in Russia, on the boarders of China (Igolkin et al., 2024).
- Use different colours to identify different experimental groups in the results (Figure 1 and 2).
- Point 2.4, mention which samples were taken from pigs after dissection (line 106).
- Line 94 ad libitum, should be italic.
- Too much self-citation by the authors.
Other than that I have no comments.
Author Response
Reviewer #2
In the manuscript entitled "Pathological Characteristics of the Emerging Recombinant African Swine Fever Virus Genotypes I and II in Vietnam", Dr. Nguyen et al., describe the pathogenesis and biological properties if ASFV recombinant strain (I/II genotype) VNUA/rASFV/HD1/23. The authors provided a good introduction, with adequate methods and results that cover all the biological properties of the strain. This work gives a good insight on the new recombinant strains that are circulating in Asia and Russia, which should be taken into consideration when developing new vaccines.
We truly appreciate your insightful comments, recommendations, and suggestions for this manuscript. We have answered your comments and concerns, and all changed are highlighted in yellow in the revised manuscript.
I have some comments.
- Reviewer’s Comment: In the introduction (line 48) and discussion, please mention the recombinant virus that was also detected in Russia, on the borders of China (Igolkin et al., 2024).
Author’s answer: The following sentence has been added (Line 49-52): “Notably, similar recombinant ASFV strains were detected in domestic pigs in Russia near the Chinese border in 2023, highlighting the potential for cross-border transmission and the increasing genetic complexity of circulating ASFV isolates [8].”
- Reviewer’s Comment: Use different colours to identify different experimental groups in the results (Figures 1 and 2).
Author’s answer: Thank you for your valuable suggestion. While we understand the importance of clear group differentiation, we would like to note that Figures 1 and 2 already employ distinct marker shapes (e.g., squares, circles, triangles) to represent different experimental groups. This approach ensures clear visual separation, including in grayscale print and for readers with color vision deficiencies. Therefore, we believe the current figure design effectively communicates the group distinctions without requiring additional color coding.
3. Reviewer’s Comment: Point 2.4, mention which samples were taken from pigs after dissection (line 106)
Author’s answer:
Original text (Line 105-106): On all dead pigs, detailed necropsies were performed immediately, and samples were taken for ASFV genome detection and histopathology.
Edited text (Line 108-111): “Extensive necropsies were immediately performed on all deceased pigs, and blood and tissue samples were taken from organs such as spleen, liver, kidneys, tonsils, and lymph nodes for ASFV genome detection and histopathological examination”.
- Reviewer’s Comment: Line 94 ad libitum, should be italic.
Author’s answer: Thank you for your comment. The term “ad libitum” has been italicized as suggested (Line 97).
- Reviewer’s Comment: Too much self-citation by the authors.
Author’s answer: Thank you for your insightful comment on the number of self-citations. We fully recognize the importance of balanced citation practices to ensure objectivity and academic integrity. The self-citations included in the manuscript were carefully selected on the basis of their direct relevance to the present study, particularly in terms of providing essential background information, methodological details, and fundamental data not readily available from other sources. We have thoroughly re-evaluated all references and confirm that each self-citation is scientifically justified and makes a meaningful contribution to the content and context of the manuscript.
Reviewer 3 Report
Comments and Suggestions for Authors
This study provides a comprehensive investigation of the pathogenicity and pathological characteristics of the emerging recombinant African swine fever virus (rASFV) genotype I/II strain (VNUA/rASFV/HD1/23) in Vietnam. The experimental design is robust, and the results are well-presented, contributing valuable insights into the virulence of this novel strain. However, several aspects require clarification or improvement to enhance the manuscript's impact and clarity.
- The dose-dependent progression of clinical signs (fever, anorexia, mortality) is clearly demonstrated. However, the statistical significance of survival time differences between groups (10², 10³, 10⁴ HAD₅₀) should be explicitly tested (e.g., log-rank test) to strengthen the conclusions. The histopathological descriptions (e.g., splenic necrosis, lymphoid depletion) align with typical acute ASFV pathology. High-resolution images of key lesions (spleen, lymph nodes) with scale bars would improve reproducibility.
- The earlier detection of ASFV in oral fluids (2 dpi in Group 3) compared to blood (3 dpi) is intriguing. Could this reflect localized replication at the inoculation site? A brief discussion on potential mechanisms (e.g., mucosal replication) would be valuable.
- The claim that VNUA/rASFV/HD1/23 exhibits "comparable pathogenicity" to other global strains (e.g., Georgia 2007/1, Pig/HLJ/18) relies on descriptive comparisons. Include quantitative metrics (e.g., mean time to death ± SD) alongside references to prior studies for direct comparison.
- The statement that rASFV I/II strains resist licensed vaccines in Vietnam (lines 56–57, 240–241) is critical but lacks experimental evidence in this study. Cite specific data or propose future work to test vaccine cross-protection against this strain.
- Table 1: Define "Ct value" in the footnote (viremia column). Specify the PCR detection limit.
- Figure 3: Label the y-axis as "Survival probability (%)" for clarity.
- Expand on the ecological implications of rASFV I/II emergence (e.g., recombination mechanisms, potential wildlife reservoirs).
Author Response
Reviewer #3
This study provides a comprehensive investigation of the pathogenicity and pathological characteristics of the emerging recombinant African swine fever virus (rASFV) genotype I/II strain (VNUA/rASFV/HD1/23) in Vietnam. The experimental design is robust, and the results are well-presented, contributing valuable insights into the virulence of this novel strain. However, several aspects require clarification or improvement to enhance the manuscript's impact and clarity.
- Reviewer’s Comment: The dose-dependent progression of clinical signs (fever, anorexia, mortality) is clearly demonstrated. However, the statistical significance of survival time differences between groups (10², 10³, 10⁴ HAD₅₀) should be explicitly tested (e.g., log-rank test) to strengthen the conclusions.
Author’s answer: We thank the reviewer for this insightful comment. The following sentence has been added (Lines 152–158): “To statistically analyze the survival differences, a log-rank test (Mantel-Cox) was performed to compare the survival curves of all four test groups. The analysis revealed a significant difference in the probability of survival (χ² = 20.6, df = 3, p = 0.0001). The control group had no mortality, while group 3 (10⁴ HAD₅₀) had the fastest and most complete mortality, which contributed significantly to the overall test statistic. These results confirm that the survival rate was significantly influenced by the infectious ASFV dose”.
- Reviewer’s Comment: The earlier detection of ASFV in oral fluids (2 dpi in Group 3) compared to blood (3 dpi) is intriguing. Could this reflect localized replication at the inoculation site? A brief discussion on potential mechanisms (e.g., mucosal replication) would be valuable.
Author’s answer: Thank you for this insightful comment. We agree that the earlier detection of ASFV in oral fluids (2 dpi) compared to blood (3 dpi) is a remarkable result. In our study, blood samples were taken at fixed intervals, every three days after inoculation, to minimize stress to the animals. Therefore, it is possible that viraemia occurred earlier than 3 days after inoculation but was not recorded due to the sampling schedule. If the blood samples had been taken daily, earlier detection in the blood would have been possible. This limitation was discussed in the manuscript (lines 271–276).
- Reviewer’s Comment: The claim that VNUA/rASFV/HD1/23 exhibits "comparable pathogenicity" to other global strains (e.g., Georgia 2007/1, Pig/HLJ/18) relies on descriptive comparisons. Include quantitative metrics (e.g., mean time to death ± SD) alongside references to prior studies for direct comparison.
Author’s answer: We thank the reviewer for his insightful commentary. The comparison has already been presented in lines 304 to 309 of the manuscript, together with the corresponding references.
- Reviewer’s Comment: The statement that rASFV I/II strains resist licensed vaccines in Vietnam (lines 56–57, 240–241) is critical but lacks experimental evidence in this study. Cite specific data or propose future work to test vaccine cross-protection against this strain.
Author’s answer: We thank the reviewer for this important comment. As noted, the manuscript addresses the potential resistance of rASFV I/II strains to licensed vaccines in Vietnam. Although this study did not include direct experimental evaluation of vaccine efficacy against the rASFV I/II strain, we have cited the recent study by Diep, N.V., et al. (2024), which presents relevant experimental data demonstrating that currently available vaccines fail to confer protection against recombinant ASFV strains circulating in Vietnam.
- Reviewer’s Comment: Table 1: Define "Ct value" in the footnote (viremia column). Specify the PCR detection limit.
Author’s answer: We thank the reviewer for this helpful comment. The following sentence has been added (Lines 161-163): “Note: Ct (cycle threshold) refers to the number of PCR cycles required for the fluorescence signal to exceed the background value. Interpretation of the Ct value: Ct < 38: Positive; 38 ≤ Ct < 40: Suspicious; Ct ≥ 40: Negative.”
- Reviewer’s Comment: Figure 3: Label the y-axis as "Survival probability (%)" for clarity.
Author’s answer: We appreciate the reviewer’s suggestion. As recommended, we have updated the y-axis label in Figure 3 to “Survival probability (%)” to improve clarity and ensure consistency with the presented data.
- Reviewer’s Comment: Expand on the ecological implications of rASFV I/II emergence (e.g., recombination mechanisms, potential wildlife reservoirs).
Author’s answer: The main objective of our study was to evaluate the virulence and pathogenicity of the recombinant ASF virus strain VNUA/rASFV/HD1/23 by experimental infection of domestic pigs. We are aware of the ecological significance of the emergence of rASFV I/II, including recombination mechanisms and the possible role of wildlife reservoirs, but these aspects were beyond the scope of the present study. Nevertheless, we agree that such ecological factors are crucial for understanding the evolution and epidemiology of ASFV. We will consider these important dimensions in future research aimed at understanding the broader impact of recombinant ASFV strains in endemic areas.
Round 2
Reviewer 3 Report
Comments and Suggestions for Authors
After the revision, the quality of the manuscript has improved significantly.
Author Response
We thank the reviewer for the positive feedback on our revised manuscript.